# Identification of Functional Cortical Plasticity in Children with Cerebral Palsy Associated to Robotic-Assisted Gait Training: An fNIRS Study

**DOI:** 10.3390/jcm11226790

**Published:** 2022-11-16

**Authors:** David Perpetuini, Emanuele Francesco Russo, Daniela Cardone, Roberta Palmieri, Chiara Filippini, Michele Tritto, Federica Pellicano, Grazia Pia De Santis, Rocco Salvatore Calabrò, Arcangelo Merla, Serena Filoni

**Affiliations:** 1Department of Neuroscience and Imaging, Institute for Advanced Biomedical Technologies, University G. D’Annunzio of Chieti-Pescara, 66100 Chieti, Italy; 2Padre Pio Foundation and Rehabilitation Centers, San Giovanni Rotondo, Viale Cappuccini 77, 71013 San Giovanni Rotondo, Italy; 3Department of Engineering and Geology, University G. D’Annunzio of Chieti-Pescara, 65127 Pescara, Italy; 4Department of Basic Medical Sciences, Neurosciences and Sense Organs, Institute of Child Neuropsychiatry, University of Bari, 70122 Bari, Italy; 5Next2U s.r.l., 65127 Pescara, Italy; 6IRCCS Centro Neurolesi “Bonino-Pulejo”, 98124 Messina, Italy

**Keywords:** cerebral palsy, neurological disorders, robotic-assisted gait training (RAGT), near infrared spectroscopy (fNIRS), machine learning (ML), rehabilitation

## Abstract

Cerebral palsy (CP) is a non-progressive neurologic condition that causes gait limitations, spasticity, and impaired balance and coordination. Robotic-assisted gait training (RAGT) has become a common rehabilitation tool employed to improve the gait pattern of people with neurological impairments. However, few studies have demonstrated the effectiveness of RAGT in children with CP and its neurological effects through portable neuroimaging techniques, such as functional near-infrared spectroscopy (fNIRS). The aim of the study is to evaluate the neurophysiological processes elicited by RAGT in children with CP through fNIRS, which was acquired during three sessions in one month. The repeated measure ANOVA was applied to the β-values delivered by the General Linear Model (GLM) analysis used for fNIRS data analysis, showing significant differences in the activation of both prefrontal cortex (F (1.652, 6.606) = 7.638; *p* = 0.022), and sensorimotor cortex (F (1.294, 5.175) = 11.92; *p* = 0.014) during the different RAGT sessions. In addition, a cross-validated Machine Learning (ML) framework was implemented to estimate the gross motor function measure (GMFM-88) from the GLM β-values, obtaining an estimation with a correlation coefficient r = 0.78. This approach can be used to tailor clinical treatment to each child, improving the effectiveness of rehabilitation for children with CP.

## 1. Introduction

Cerebral palsy (CP) is a non-progressive neurologic condition arising from a brain injury that occurs before cerebral development is complete [1], and it is a leading cause of motor disability in children. In fact, it is estimated that the average incidence of CP is 2.08 per 1000 live births, but when only children born with a body weight below 1500 g are considered, the incidence increases 70-fold compared to children born with a body weight over 2500 g. [2]. CP provokes muscle weakness, spasticity, bone deformities and impaired balance and coordination. Children with CP often exhibit a decrease in walking speed, an increase in double support duration, and poor endurance, with an impairment in daily activities, community integration and quality of life [1]. the Gross Motor Function Classification System (GMFCS) is one of the most diffuse and easy-to-use scales to describe the severity of motor disorders of CP [3]. This scale evaluates the child’s independence when performing basic motor functions, and classifies the patients into five classes, where class 1 includes individuals who can freely walk, whereas class 5 includes subjects who are not able to move on their own [3]. 

Several pharmacological approaches [4] and rehabilitative techniques have been developed [5] to improve the quality of life of CP patients and their social integration. Robotic-assisted gait training (RAGT) has become a common rehabilitation tool employed to improve the gait pattern of people with neurological impairments [6]. In fact, it was demonstrated that passive movements of the limbs could activate the central pattern generators (CPG), i.e., gait centers in the spinal cord [7]. Importantly, it was proved that passive activation of the legs on a treadmill could evoke similar locomotor activity also in patients with severe spinal cord injuries [6]. With respect to the conventional body weight support (BWS) treadmill training methods, RAGT has many advantages. For instance, RAGT allows the early beginning of rehabilitation also in severe patients, administering higher intensity of training as compared to traditional rehabilitation. Moreover, RAGT forces the patients into more physiological and reproducible gait patterns, providing the possibility to evaluate the patient’s performances. Finally, RAGT produces benefits also for cardiopulmonary functionalities [8].

Two typologies of RAGT devices are commercially available: end-effectors and exoskeletons. End-effectors are basically composed of a double crank and rocker gear system to apply forces to the distal segments of limbs, and the patient is guided during the gait by a servo-controlled motor. Exoskeleton-type devices are robotic orthosis combined with a harness-supported body weight system combined with a treadmill, and the patient’s legs are guided by the robotic device following a preprogrammed gait pattern. Among the exoskeletons commercially available, the Lokomat^®^ (Hocoma AG, Volketswil, Switzerland) is a device that supports the patient on a treadmill while a robotic apparatus facilitates the inter-limb coordination and gait timing. The device provides several degrees of BWS and guidance, that could be modulated according to the patient’s needs, resulting in highly suitable and effective to treat CP children [9]. In fact, it was demonstrated that the employment of the pediatric orthosis tool could significantly decrease muscle stiffness after a single session of RAGT [9]. 

Moreover, it is worth noting that RAGT can produce a higher spinal and brain neuroplasticity as compared to the conventional BWS treadmill training methods [6]. Neuroplasticity is the ability of the central nervous system (CNS) to go through permanent structural and functional modification in response to internal and external stimuli [10]. This capability can be exploited in a damaged brain, which adapts its functionalities in response to rehabilitation. Specifically, CP patients’ brains could be able to recover the control of motor functions through the neuroplasticity mechanism induced by rehabilitation. Notably, CNS exhibits a higher plasticity at the earliest stages of its development; hence, rehabilitation of children with CP should be started early [5,11]. 

Motor rehabilitation permits the restoration of lost motor patterns or the development of new patterns that can compensate for irretrievably lost functions by means of compelled motor activity. In this perspective, measuring brain activity during the RAGT could provide information about the effect of the therapy on neuroplasticity. To this aim, portable neuroimaging techniques are more suitable than the techniques requiring high constraints for the patients (e.g., functional Magnetic resonance, fMRI, magnetoencephalography, MEG) [12]. Particularly, functional near-infrared spectroscopy (fNIRS) is an optical neuroimaging technique able to measure the variations of oxyhemoglobin (HbO) and deoxyhemoglobin (HHb) secondary to the cortical activations, exploiting the optical properties of these chromophores in the near-infrared spectral range [13,14]. Hence, in similarity with functional Magnetic Resonance (fMRI), fNIRS is able to measure neuronal activation through the Blood Oxygen Level Dependent (BOLD) effect. Moreover, the feasibility of the combination between fNIRS and motor rehabilitation in CP children has been already demonstrated [15], and the capability of fNIRS to evaluate functional plasticity for several rehabilitation purposes has been already explored [16,17]. 

The aim of this study was to evaluate the neurophysiological processes activated by RAGT, in addition to conventional therapy, using fNIRS in children with CP. The novelty of this study relies on the demonstration of the possibility to employ fNIRS during RAGT in children with CP to evaluate in vivo the neuroplasticity induced by the rehabilitation. 

## 2. Materials and Methods

### 2.1. Particpants

Ten children with CP attending the Neurorehabilitation Unit “Gli Angeli di Padre Pio” of the Padre Pio Foundation and Rehabilitation Centers, San Giovanni Rotondo, Foggia, Italy, between November 2021 and April 2022 were enrolled in this study. Two of them dropped out as they were positive for SARS-CoV-2 during the training, so the final sample was composed of eight individuals. Table 1 summarizes the participants’ characteristics.

Inclusion criteria were: (1) children with CP aged between three and eighteen with a GMFCS level of I-V, (2) ability to communicate discomfort or pain, (3) understanding simple instructions.

Exclusion criteria included medical conditions potentially interfering with the locomotor training and physical restrictions for using the robotic device. In addition to all criteria defined in the Lokomat manufacturer’s manual, children were excluded if they exhibited: severe lower-extremity muscle contractures, hip instability/subluxation, Botulinum toxin-A (BTX-A) injections to lower limbs within the last 3 months, uncontrolled seizure disorder, open skin lesions, or vascular disorder of the lower limbs; or if patients were unable to cooperate or be positioned appropriately within the Lokomat.2.2 Clinical evaluation.

The participants underwent 3 RAGT sessions per week, each session lasting about 30 min, for a total of 12 RAGT sessions. RAGT was administered with the biofeedback provided by the Lokomat. In addition to the RAGT, the participants also received traditional therapy. In order to provide an initial evaluation of the motor abilities of the participant, the Gross Motor Function Classification System (GMFCS) was employed. It provides a method for describing the functional ability of children with CP at one of the five levels. Children in Level I can perform all the activities of their age-matched peers, albeit with some difficulty with speed, balance, and coordination; children in Level V have difficulty controlling their head and trunk posture in most positions and in achieving any voluntary control of movement [18].

In order to evaluate the modifications of the motor abilities of the participants, the following clinical scales were administered before (T0) and at the end of the training (T2):Gross Motor Function Measure-88 (GMFM-88): it consists of 88 items in five dimensions: lying and rolling (GMFM-A); sitting (GMFM-B); crawling and kneeling (GMFM-C); standing (GMFM-D); and walking, running and jumping (GMFM-E) [19].Modified Ashworth Scale (MAS): it was developed by Bryan Ashworth as a method of grading spasticity. The original Ashworth scale was a 5 points numerical scale that graded spasticity from 0 to 4, with 0 being no resistance and 4 being a limb rigid in flexion or extension. However, Bohannon and Smith modified the Ashworth scale by adding 1+ to the scale to increase sensitivity. Hence, MAS varies from 0 (no increase in muscle tone) to 4 (affected part(s) rigid in flexion or extension) [20].

Shapiro–Wilk’s normality test was applied to check the normality of the clinical data distribution of the scores, for both the GMFM-88 and the MAS. Since the data did not meet the assumption of normality, the Wilcoxon signed-rank test was performed between the scores obtained from the the GMFM-88 and the MAS during the different sessions (i.e., T0 and T2) to evaluate the improvement of the muscular tone and the level of spasticity in response to RAGT.

### 2.2. Experimental Procedure

After the participants were harnessed to the Lokomat, they received the RAGT experimental protocol (Figure 1a). Particularly, a block paradigm was used: the children were first asked to actively move during the RAGT for 30 s, then they were invited to rest for other 30 s. The paradigm was provided in 10 blocks, as described in Figure 1b.

### 2.3. fNIRS Measurements and Data Analysis

To measure the cortical hemodynamic activity, the portable fNIRS Cortivision Photon cap device was used (the device was provided through Cortivsion Pathfinder Program grant number CPP-2021/10/3). It is composed of 16 LEDs emitting at 760 nm and 850 nm wavelengths and 10 detectors. The montage delivered 34 channels covering the frontal, prefrontal and motor cortices, placing the optodes according to the 10–20 system. Moreover, 4 short channels have been provided in order to remove the physiological contaminations from the fNIRS signals [21]. The fNIRS was recorded at the first session (T0), at the sixth (T1) and at the twelfth session (T2).

The locations of the sources and detectors were digitized using a Polhemus FastTrak 3D digitizer (Colchester, Vermont, United States; accuracy: 0.8 mm) comprised of a recording stylus and three head-mounted receivers employed for small head movements between acquisitions. Figure 2 depicts the average channel locations among subjects warped into MNI space (Colin27). The Brodmann Areas (BAs) investigated were 1, 4, 5, 6, 7, 8, 9, 10, 11, 40, 45, and 46, according to a sensitivity analysis performed in NIRS-SPM [22].

The fNIRS signals were converted into optical densities and then converted into HbO and HHb concentrations through the Modified-Lambert-Beer law. The DPF was computed for each participant in accordance with Scholkmann et al [23] and Chiarelli et al [24]. The hemoglobin signals were filtered with a zero-lag 3rd order Butterworth digital filter (cut-off frequencies 0.01–0.4 Hz) [25] and the motion artifacts were removed through a Wavelet transform based algorithm [26]. Physiological contaminations have been corrected from the long separation channels through the procedure reported by Sato and colleagues [27]. 

The fNIRS data analysis was based on General Linear Model (GLM), which, in the matrix notation, is expressed as:
Y = Xβ + ε(1)where Y is a n × 1 column vector indicative of the considered time-series; X is a n × p design matrix (n is the number of rows and p the number of columns of the design matrix), where the different columns correspond to a predictor variable; β is a p × 1 column vector of predictor weights indicating the strength of the relation with Y.; and ε is an n × 1 column vector associated to the residual error. The undisclosed values of β is evaluated using least-squares regression, which provides information regarding the amount of signal variance explained by the predictor. [28]. For fNIRS applications, the β-values are indeed indicative of the cortical activation and could be used for further statistical analysis. In order to assess statistical differences between the β-values at T0, T1 and T2, a one-way repeated measure ANOVA (RM-ANOVA) has been performed for each channel, where the within factor is the temporal session. In order multiple comparisons (paired *t*-test) were performed to assess which temporal recordings provided statistical differences, and, the results were Bonferroni corrected to avoid false positives, considering the number of channels and comparisons. This analysis was performed for both HbO and HHb.

Finally, a machine learning (ML) regression analysis based on Gaussian Process Regression was performed linking the cortical activations with the GMFM-88, to test whether the assessed changes in the brain activity were associated with modifications in the motor abilities. In detail, the β-values at T0 and T2 obtained from the GLM analysis (for both HbO and HHb) were used as input, whereas the GMFM-88 was used as output. Of note, a feature selection based on the Wrapper procedure [29] was performed to reduce eventual overfitting effects. A leave-one-subject-out cross validation was employed to assess the generalization performances of the ML framework.

## 3. Results

### 3.1. Clinical Scales Results

The results of the comparison of the clinical scales evaluated before (T0) and after the RAGT sessions (T2) are reported in Table 2. In particular, only the GMFM-88 showed a significant difference between T0 and T2 (T0 vs. T2, z = −2.524; *p* = 0.008), as shown in Figure 3. 

The variation of the GMFM-88 scores between T0 and T2 and the GMFCS level for each participant is reported in Table 3.

### 3.2. fNIRS Results

Figure 4a reports the F-statistic map obtained performing the RM-ANOVA between T0, T1 and T2 for HbO signals, whereas Figure 4b reports the map obtained through the same comparison for the HHb signal. Notably, the figure highlights the channels that were still significant after the correction. Specifically, BA 9 and BA 11 showed significant differences bilaterally, whereas BA 6 and 45 showed a significant difference only for the left hemisphere, and BA 1 and 46 for the right one.

The significant post-hoc multiple comparison results are reported in Table 4. Of note, significant differences, applying the Bonferroni correction, were obtained only between T0 and T2, except for the left frontal zone (corrected *p*-value: 0.04), which also showed differences between T0 and T1. It is worth highlighting that differently to the GMFM-88, the fNIRS was acquired at T0, T1 and T2, hence the RM-ANOVA was employed to assess differences between the three temporal points and the post-hoc analysis was performed between all three measurement times considered.

Finally, the regression analysis demonstrated an association between the cortical activation modifications and the improvement of the motor abilities. In detail, the Wrapper procedure selected channels 7, 12 and 14. This subset of features was used as input of the Gaussian Process Regression, which estimated the GMFM-88 with a correlation coefficient of 0.78 (*p* = 0.001) and a root mean square error of 0.636, as reported in Figure 5.

## 4. Discussion

The purpose of this study was to evaluate through fNIRS the neurophysiological processes activated by RAGT in children with CP. The results revealed bilateral changes in the cortical activations of BA 1, 6, 9, 11 and 46. In particular, the post-hoc analysis revealed a decrease in BA 9 activation during the session and an increase in BA 1, 6, 11, and 46 activities. Specifically, BA regions 11, 6, and 9 are involved in motor movement sequence planning, reward, long-term memory, and decision-making. In addition, BA 6 and 9 are involved in the control of movement intention, complex movements, and coordination. BA 46 and 9 are also associated with attention, working memory, and self-control, whereas BA 1 is associated with proprioceptive and fractional movement control. Finally, BA 9 is also associated with spatial memory [30]. Notably, the multiple comparisons analysis delivered significant differences, Bonferroni corrected, only between T0 and T2, except for the left BA 46, which shows differences also between T0 and T1. Importantly, these modifications in cortical activity are associated with improvements in the motor functions, as shown by the significant increase in the GMFM-88 and sub-scores. This finding is in line with previous studies demonstrating the benefits of the RAGT in diplegic children with CP. For instance, Wallard et al found a significant improvement of the kinematic data of the full-body in the sagittal and frontal planes and of the Gross Motor Function Measure test due to the RAGT therapy administered through Hocoma Lokomat, demonstrating the usefulness of RAGT in improving the balance control in gait [31]. Van Kammen and colleagues found positive effects associated to the RAGT as well. In particular, they demonstrated that walking with the Lokomat reduces hypertonia in children with CP; whereas altering guidance or BWS generally does not affect amplitude [32]. Nonetheless, these studies do not provide information regarding cerebral functioning modifications associated to RAGT and motor control improvements. In this perspective, it should be highlighted that several neurological diseases are associated with gait disorders whose neurofunctional correlates are scarcely investigated. To this aim, fNIRS is particularly suitable to study neurophysiological modifications during rehabilitation since it allows investigating functional brain hemodynamic oscillations in ecological settings, without hard constraints for the individual [12] and the present work constitutes a fundamental step forward the investigation of this neurophysiological aspects. Notably, the scores of every participant on the GMFM-88 increased. Specifically, the greatest changes were associated with the lowest GMFCS levels. This result may indicate that RAGT is more effective in patients with severely impaired motor abilities. Since it is known that different GMFCS levels are characterized by different cortical activations during motor tasks [33], to gain more insight into the relationship between the baseline motor abilities of the patients and the efficacy of the RAGT, also in terms of neural plasticity, it is necessary to conduct additional research involving a larger number of patients with varying GMFCS levels.

Concerning the employment of fNIRS during gait, Kurz et al. found that children with CP exhibit increased activation in the sensorimotor cortices and superior parietal lobule during gait with respect to typical developed children. Moreover, they demonstrated that children with CP had a higher variability and number of errors in stride time intervals and temporal gait kinematics [15]. Although Kurz et al evaluated the cortical activity during walking on a treadmill (and not during RAGT), their findings could provide insights in the interpretation of the results of our study. In fact, the multiple comparison analysis showed a decreased cortical activation through the RAGT sessions in the BA 9, which is related to the motor planning and coordination, suggesting that this robotic training could help the brain to function in a manner similar to the typical developed children, and this is characterized by a reduced brain activity in the motor cortex. Moreover, Chaudary and colleagues observed a bilateral dominance in the prefrontal cortex of healthy controls and an ipsilateral dominance in CP patients during a ball throwing task [34]. Concerning the use of fNIRS during RAGT, Van Hedel et al. evaluated the cortical activity through fNIRS in children with neurological disorders during walking both with the Andago (a device enabling over-ground walking with bodyweight support) and on a treadmill. They demonstrated that only a small proportion of the participants with neurological disorders show typical hemodynamic responses during walking in Andago, in contrast with healthy controls. Moreover, they demonstrated high levels of acceptance of the fNIRS by children with neurological disorders during the motor rehabilitation.

The increased activation found in the prefrontal cortex could be associated to an increase in the attention and participation of the patients during the RAGT session. It is worth highlighting that increased activity in prefrontal cortex is associated with a rise in cognitive load. However, an increase in prefrontal cortex activation is not always associated with an excessive workload [35]. The improvement in motor performance across sessions assessed in this study suggests that the evaluated increase in prefrontal activity is positively related to concentration, attention, and engagement with therapy. This hypothesis could be corroborated by the ML analysis. In fact, the developed ML framework evaluated the relationship between the hemoglobin oscillations and the modifications in the motor functions (GMFM-88). In order to reduce the risk of an overfitting effect, an automated features selection relying on the Wrapper approach was conducted. The procedure selected channels that cover the prefrontal and somatosensory cortex, showing a link between cortical activity modifications in those areas and the motor functions improvements due to RAGT. Specifically, the Wrapper procedure selected channels that cover the prefrontal and somatosensory cortexes. Particularly, channels 7, 12, and 14 were selected. Channels 7 and 12 covered BA 46, whereas Channel 14 covered BA 9. The post-hoc analysis showed that channel 7 was not significant, channel 12 was significant for both the contrasts T0 vs. T1 and T0 vs. T2, showing increased activity across the sessions, and channel 14 showed a significant difference between T0 and T2, exhibiting decreased activity across sessions. These results seem to suggest that the improvements in the motor abilities assessed through the GMFM-88 are associated with increased attention and self-control and decreased motor movement sequence planning, control of movement intention, complex movements, and coordination. However, it should be highlighted that further studies are indeed necessary to increase the sample size, in order to further corroborate the generalization of the results. In fact, since the fNIRS-based regression outcome relies on a multivariate analysis of all the channels, the regression performances might strongly increase enlarging the sample numerosity. However, it should be highlighted that, although the sample size could be considered small, the ML regression was performed using a leave-one-out cross-validation technique, which excludes one participant at a time and tests the classifier on that participant, thereby evaluating the out-of-sample performance and delivering generalizable results. These results could pave the way to the employment of the fNIRS in clinical practice with the aim of estimating the responsiveness of the children to the therapy in terms of neural plasticity and motor functions, allowing to modulate the treatment to the needs of the patients, thus increasing its effectiveness. These findings could also foster the establishment of shared guidelines regarding the administration of the RAGT. In fact, there are no standardized dosage and guidelines of the RAGT in CP. In fact, there are only two reviews describing respectively 486 CP patients in 17 studies [36] and 217 patients in 10 studies [37] undergoing robotic treatment. The data from these works show that the majority of the studies focus more on children with CP classified as I-IV GMFCS level I to IV, excluding more serious patients with GMFCS V level. Moreover, heterogeneity in the choice of treatment protocol, whose duration varies from a minimum of 30 min to a maximum of 60 min, is described by the two reviews. Moreover, sessions range from 2 to 5 per week, repeated for 2-6 weeks, up to a maximum of 10 weeks as reported by Sarhan’s et al. [38]. Then, we suggest employing fNIRS in the clinical setting to standardize the RAGT protocols in children with CP, in order to avoid the heterogeneity of the clinical approach and maximize the neurological and motor benefits of the therapy.

It is noteworthy that our study shows that most of the changes in brain activity are found between T0 and T2, suggesting that a minimum number of sessions is needed to produce brain plasticity effects. However, it should be highlighted that the protocol was carried out 3 times a week for a total of 12 sessions. It would be interesting to verify if the same changes in the cortex would have occurred in intensive daily training. Moreover, it should be highlighted that the present study was limited to a protocol of 12 sessions, but a medium and long-term follow-up would be necessary to investigate whether the modification of the cortical activity and the clinical improvements detected through GMFM 88 are definitively acquired by the children.

It should be stressed that the protocol excluded uncooperative children who were unable to follow the augmentative feedback presented on screen by the Lokomat. However, it should be investigated in future studies whether intensive RAGT could provide benefits also in non-cooperative children with severe cognitive deficits, who are normally excluded a priori from robotic rehabilitation protocols. As future perspectives, further studies comparing groups of children treated with RAGT and biofeedback with children undergoing RAGT only should be fostered. In addition, it would be interesting to compare a control group with CP children undergoing conventional gait training and RAGT. 

## 5. Conclusions

This study is the first application of neuroimaging in children with CP during RAGT, providing information on the neural substrates related to the locomotor behaviors of children with CP. The results showed that robotic therapy produces modifications in brain activity in both the motor and frontal cortex with an improvement in motor control and attention during RAGT. This approach can be used to tailor clinical treatment, improving the effectiveness of rehabilitation for children with CP. This study could pave the way for further experiments aimed at monitoring and predicting the effectiveness of rehabilitation also in other pathologies with the aim of administering the most suitable therapy for each patient.

## Figures and Tables

**Figure 1 jcm-11-06790-f001:**
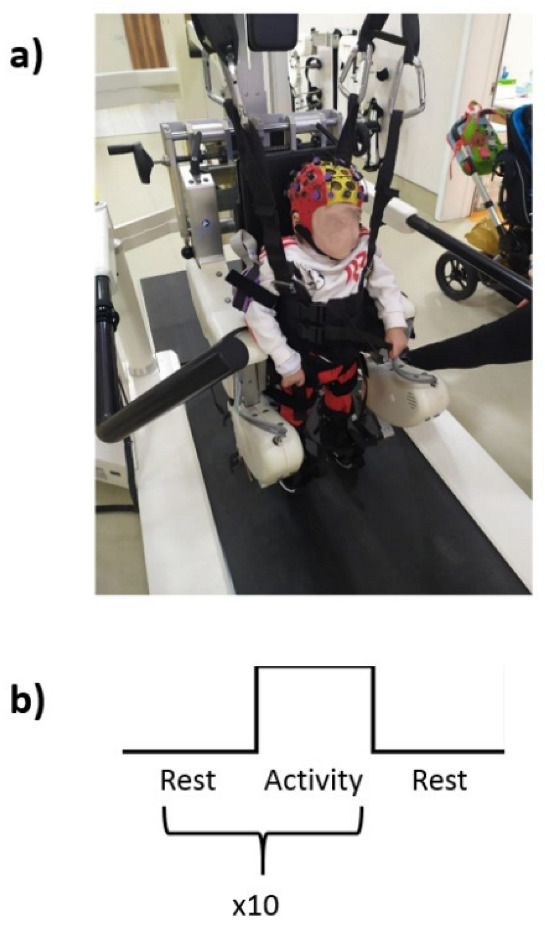
(**a**) Representative participant harnessed to the Lokomat (**b**) schematic block paradigm of the experiment.

**Figure 2 jcm-11-06790-f002:**
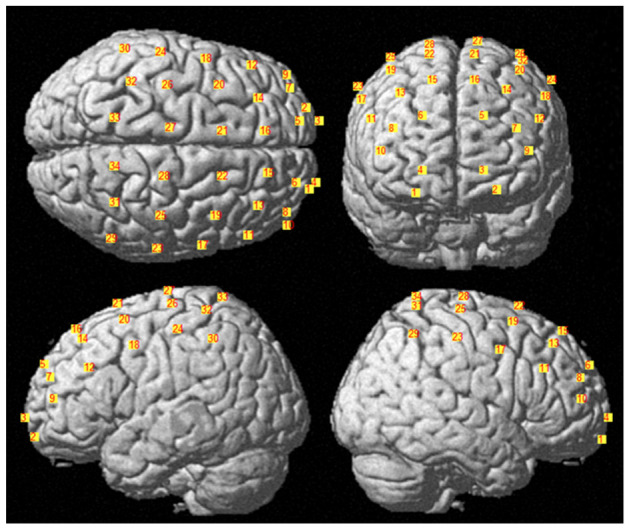
Average channels positions obtained warping the sources and detectors into MNI space. The number associated to each channels is specified in the figure.

**Figure 3 jcm-11-06790-f003:**
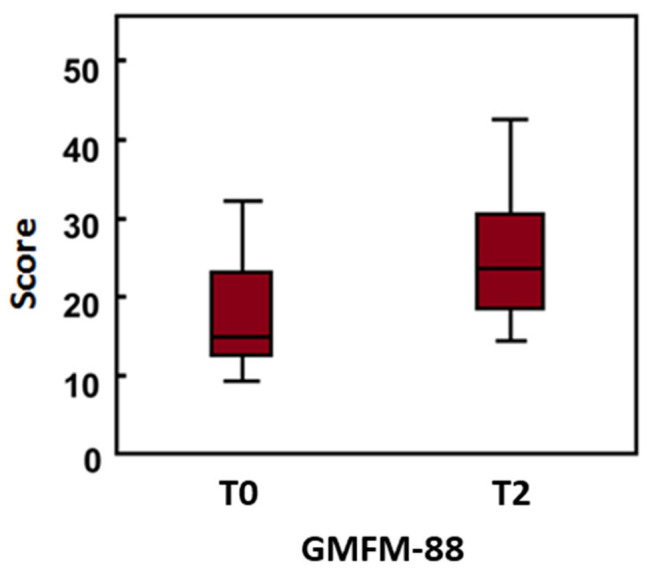
Boxplot associated to the Gross Motor Function Measure (GMFM-88) evaluated at T0 and T2.

**Figure 4 jcm-11-06790-f004:**
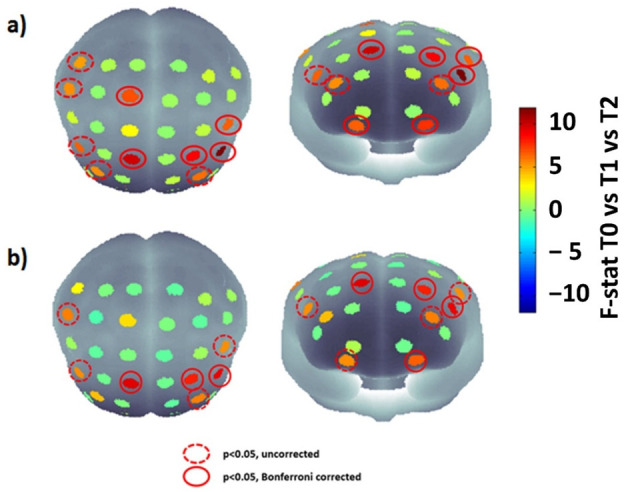
Comparative statistical maps (F-statistic) of cortical activation for HbO (**a**) and HHb (**b**) at T0, T1, and T2 showing the RM-ANOVA results. Significant channels (*p* < 0.05) that do not get through Bonferroni correction are represented by red circles with dashes, whereas continuous red circles represent channels that remain significant (*p <* 0.05) after the multiple comparison correction.

**Figure 5 jcm-11-06790-f005:**
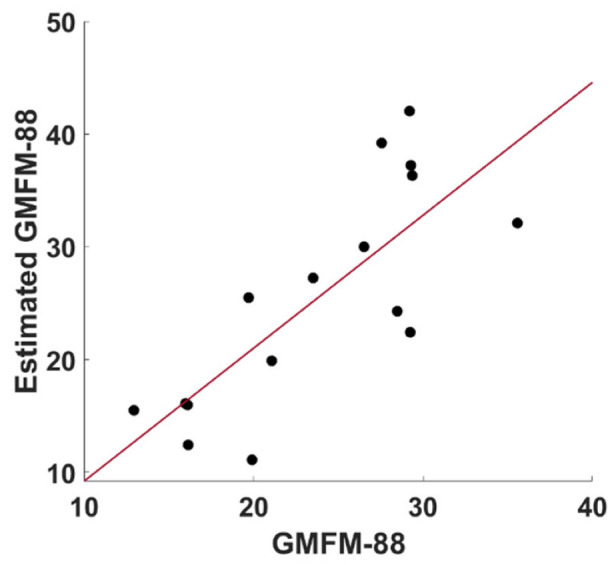
Scatter plot between the evaluated GMFM-88 and the estimated GMFM-88 obtained through a leave-one-subject-out cross-validated Gaussian Process Regression from fNIRS β-values selected employing a Wrapper procedure.

**Table 1 jcm-11-06790-t001:** Participants’ characteristics. Notably, the age is expressed as mean ± standard deviation; whereas the qualitative parameters are expressed as absolute frequencies.

**Participants**	8
**Age (years)**	9.88 ± 4.73
**Affected side (n)**	
Unilateral	1
Bilateral	7
**Cerebral palsy subtype (n)**	
Spastic	8
Dyskinetic	0
Ataxic	0
**Gross Motor Function Classification System (GMFCS) level (n)**	
I	1
II	0
III	1
IV	5
V	1

**Table 2 jcm-11-06790-t002:** Median and IQR of clinical scale at baseline (T0) and post-treatment (T2). GMFM 88, Gross Motor Function Measure-88 total values; MAS H, Modified Ashworth Scale for hip muscles; MAS K, Modified Ashworth Scale for knee muscles; MAS A, Modified Ashworth Scale for ankle muscles.

	(T0) Median (IQR)	(T2) Median (IQR)	*p* Values
GMFM 88	16 (14–23.6)	24.3 (19.4–30.9)	Z = −2.524; *p* = 0.008
MAS H	1 (0.5–2)	1 (0.75–1)	n.s.
MAS K	2 (0.5–2)	1 (0–2)	n.s.
MAS A	2 (1–2)	1 (0.75–1)	n.s.

**Table 3 jcm-11-06790-t003:** GMFM-88 improvement for each participant between T0 and T2 in response to RAGT.

Subj	Gross Motor Function Classification System (GMFCS) Level	Delta between T0 and T2 (%)
1	III	30.8
2	IV	27.0
3	I	4.1
4	IV	48.3
5	IV	21.0
6	IV	51.9
7	V	44.5
8	IV	31.4

**Table 4 jcm-11-06790-t004:** Multiple comparisons (paired *t*-test) associated with the RM-ANOVA. Only significant comparisons were reported in the table.

	Channels	Brodmann Areas	Comparison	*t*-Stat	Adjusted *p*-Value
	Ch1	1-Orbitofrontal area	T0 vs. T2	−3.747	0.020
	Ch2	1-Orbitofrontal area	T0 vs. T1	−3.799	0.018
	Ch12	46-Dorsolateral prefrontal cortex	T0 vs. T1	−3.788	0.019
Oxyhemoglobin (HbO)	Ch12	46-Dorsolateral prefrontal cortex	T0 vs. T2	−4.181	0.014
	Ch14	9-Dorsolateral prefrontal cortex	T0 vs. T2	5.937	0.004
	Ch15	9-Dorsolateral prefrontal cortex	T0 vs. T2	4.756	0.009
	Ch18	6-Pre - Motor and Supplementary Motor Cortex	T0 vs. T2	−4.075	0.015
	Ch28	4-Primary Motor Cortex	T0 vs. T2	−4.037	0.016
	Ch2	1-Orbitofrontal area	T0 vs. T1	−3.083	0.036
Deoxyhemoglobin	Ch12	46-Dorsolateral prefrontal cortex	T0 vs. T1	−3.471	0.026
(HHb)	Ch12	46-Dorsolateral prefrontal cortex	T0 vs. T2	−4.093	0.014
	Ch14	9-Dorsolateral prefrontal cortex	T0 vs. T2	5.099	0.007
	Ch15	9-Dorsolateral prefrontal cortex	T0 vs. T2	4.056	0.015

## Data Availability

The data presented in this study are available on request from the corresponding author. The data are not publicly available due to privacy issues.

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
