# Peer review of "Identification of Functional Cortical Plasticity in Children with Cerebral Palsy Associated to Robotic-Assisted Gait Training: An fNIRS Study"

_jcm, 2022, doi:10.3390/jcm11226790_

Round 1

Reviewer 1 Report

The idea and aims are new, using fNIRS as assessment tool for brain activity in CP children during applying Robotic assisstive gait training. 

The introduction is clear, if you can remove some unneeded details will be better, eg  lines 102-113 used again in the discussion.

The aim: is clear, but how can you confirm neuroplasticity by using the fNIRS? cerebral function or oxygenation is more appropriate as far I know.

Methodology: more clarifications are needed. 

revision of the equation. The name should be mentioned before the appreviations eg. n, p, FDR, BWS..

The results:

Lines 239- 246 (The results of the comparison of the clinical scales evaluated before (T0) and after  the RAGT sessions (T2) are reported in Table 2. In particular, only the GMFM-88 showed a significant difference between T0 and T1 (T0 vs T1, z=-2.524; p=0,008), as shows in  Figure 3. 

Table 2 Median and IQR of clinical scale at baseline (T0) and post-treatment (T2). GMFM 88, Gross Motor Function Measure-88 total values; MAS H, Modified Ashworth Scale for hip muscles; MAS K, Modified Ashworth Scale for knee muscles; MAS A, Modified Ashworth Scale for ankle muscles)   Table 3 represent comparison between T0 vs T1, . Figure 3 represent comparison for GMFM 88 between T0 and T2 not T1. please revise and correct, add relevent statistics to the table.

Table 3 needs more clarifications (263-266).

Figure 4 (line 278) is figure 5 as mentioned in the text (line 276).

Discussion:

Line 303 : Do you mean reduces hypertonia (Lokomat reduces muscle activity in children).

Lines 282-283 in the conclusion I think it is better to remove and, consequently,  their quality of life (This approach can be used to tailor clinical treatment, improving the effectiveness of rehabilitation for children with CP and, consequently,  their quality of life.)  same in the abstract.

Reviewer 2 Report

This is an interesting article on brain changes following robotic gait training. I have some suggestions to improve clarity and acknowledge limitations of the study.

Page 3, table 1: This is a heterogeneous sample (GMFCS I-V), which should be discussed as a potential limitation of the study because brain activity may vary based on GMFCS levels (reference below). I personally do not think children at GMFCS level I should even be placed in a robotic device for passive gait, but this is another discussion.

(Sukal-Moulton, T., de Campos, A. C., Alter, K. E., Huppert, T. J., & Damiano, D. L. (2018). Relationship between sensorimotor cortical activation as assessed by functional near infrared spectroscopy and lower extremity motor coordination in bilateral cerebral palsy. NeuroImage: Clinical20, 275-285.)

Page 4, line 1 49: As GMFCS is a classification system and not an outcome measure, I strongly recommend making clear that it was only used at baseline.

Page 4, lines 166-169

Please clarify which tests were used for the other outcome measures. The methods section only mentions using the Wilcoxon test for Ashworth scores but not the tests used for other measures. 

Figure 4: The choice of the f-stats comparting T0, T1 and T2 is unusual and hinders the understanding of the figure. I went back and forth from figure 4 to table 3 to locate the direction of the comparisons, but it is still hard to tell. I feel that the illustration should have a more straightforward reading, perhaps using the contrasts could make it easier? An orientation regarding the underlying anatomy would also be useful. 

Discussion

The first paragraph of the discussion could be improved for clarity.

Page 10, lines 326-340. I recommend adding supporting literature to the interpretation of increased activity in the prefrontal cortex as a positive aspect indicating better attention and participation. Research suggest that it may also indicate increased cognitive load. 

As the study suggests potential applications of fNIRS to estimate responsivity to the intervention,  it would be important to explore potential factors and limitations of the study with this regard – especially concerning the small sample size and variability in clinical presentations.  

The Machine learning analysis was poorly explored in the discussion. I do believe that there may be correlations of brain changes with improved GMFM, but a much deeper discussion is needed so it makes sense. For example, the channels chosen in the analysis correspond to which brain areas? Were they significantly changed after the intervention? In which direction? – ie. is increasing or decreasing the activity in these areas good for motor function?

Author Response

Please, see the attached file
